# Active Optics in Astronomy: Freeform Mirror for the MESSIER Telescope Proposal

## Gerard Rene Lemaitre *, Pascal Vola * and Eduard Muslimov * 

Laboratoire d'Astrophysique Marseille (LAM), Aix Marseille Université (AMU) and CNRS,
38 rue Frédéric Joliot-Curie, 13388 Marseille CX 13, France

* Correspondence: gerard.lemaitre@lam.fr (G.R.L.); pascal.vola@lam.fr (P.V.); eduard.muslimov@lam.fr (E.M.)

**Abstract:** Active optics techniques in astronomy provide high imaging quality. This paper is dedicated to highly deformable active optics that can generate non-axisymmetric aspheric surfaces—or freeform surfaces—by use of a minimum number of actuators. The aspheric mirror is obtained from a single uniform load that acts over the surface of a closed-form substrate whilst under axial reaction to its elliptical perimeter ring during spherical polishing. MESSIER space proposal is a wide-field low-central-obstruction folded-two-mirror-anastigmat or here called briefly three-mirror-anastigmat (TMA) telescope. The optical design is a folded reflective Schmidt. Basic telescope features are 36 cm aperture, $f/2.5$, with $1.6° \times 2.6°$ field of view and a curved field detector allowing null distortion aberration for drift-scan observations. The freeform mirror is generated by spherical stress polishing that provides super-polished freeform surfaces after elastic relaxation. Preliminary analysis required use of the optics theory of 3rd-order aberrations and elasticity theory of thin elliptical plates. Final cross-optimizations were carried out with Zemax raytracing code and Nastran FEA elasticity code in order to determine the complete geometry of a glass ceramic Zerodur deformable substrate.

**Keywords:** telescope; active optics; aspherics; elasticity; optical design; deformable mirror

## 1. Introduction

Technological advances in astronomical instrumentation during the second part of the twentieth century gave rise to 4 m, 8 m and 10 m class telescopes that were completed with close loop *active optics* control in monolithic mirrors [1], segmented primary mirrors [2] and segmented in-situ stressed active optics mirrors [3]. Further advances on ground-based telescopes allowed developments of *adaptive optics* for blurring the image degradation due to atmosphere. Both *active and adaptive optics* provided milestone progress in the detection of super-massive black holes, exoplanets and large-scale structure of galaxies.

This paper is dedicated to *highly deformable active optics* that can generate non-axisymmetric aspheric surfaces—or *freeform surfaces*—by use of a minimum number of actuators: a single uniform load acts over the monolithic surface of a *closed-form* substrate whilst under reaction to its elliptical perimeter ring. These freeform surfaces are the basic aspheric components of dispersive/reflective Schmidt systems. Both reflective diffraction freeform gratings and freeform mirrors have been investigated by Lemaitre [4].

MESSIER proposal is a wide-field low-central-obstruction folded-two-mirror-anastigmat or here called briefly three-mirror-anastigmat (TMA) telescope based on a folded reflective Schmidt optical design. It is dedicated to the survey of extended astronomical objects with extremely low surface brightness. The proposal name is in homage to French astronomer Charles Messier (1730–1817) (Figure 1) who published the first astronomical catalogue of nebulae and star clusters. The catalogue was known as *Messier objects*—for instance, M1 for the Crab nebulae, M31 for the Andromeda galaxy

and so forth—and distinguished between permanent and transient diffuse objects, then helped in sensing or discovering comets.

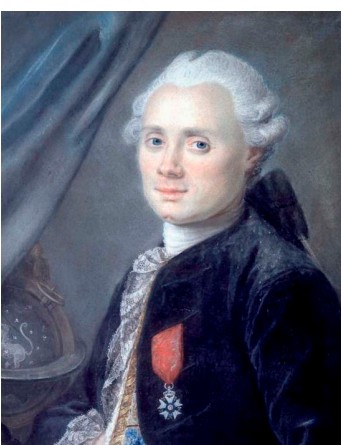

**Figure 1.** Charles Messier (1730–1817) (Wikipedia).

MESSIER optical design would provide a high imaging quality without any diffractive spider pattern in the beams. A preliminarily ground-based prototype is planned to serve as a fast-track pathfinder for a future space-based MESSIER mission. The elliptical freeform mirror is generated by stress polishing and elastic relaxation technique [4,5].

Super-polishing of freeform surfaces is obtained from spherical polishing after elastic relaxation. Preliminary analysis required use of the optics theory of 3rd-order aberrations and elasticity theory of thin elliptical plates. The final cross-optimizations were carried out with Zemax ray-tracing code and Nastran FEA elasticity code that provides accurate determination of the substrate geometry of a glass ceramic Zerodur material.

## 2. Optical Design with a Reflective Schmidt Concept

A reflective Schmidt design is a wide-field system and basically a two-mirror anastigmat. With one freeform surface correcting all three primary aberrations, such a system has been widely investigated as well as for telescope or spectrograph designs. The corrector mirror or reflective diffraction grating is always located at the centre of curvature of a spherical concave mirror.

Compared to a Schmidt with a refractor-correcting element, which is a centred-system, a reflective Schmidt necessarily requires a tilt of the optics. The inclination of the mirrors then forms a *non-centred-system*. For an $f/2.5$ focal-ratio, the tilt angle of the aspheric primary mirror is typically of about $10°$ and somewhat depending on the FoV size.

For a reflective Schmidt optical design, as MESSIER telescope proposal at $f/2.5$, a supplementary folding-flat mirror was found necessary. This third mirror provides: (i) an easy access to the detector for minimal central obstruction; and (ii) avoid any spider inside the beams that would cause diffractive artefact cross-like images at the focal image of bright stars. This led us to a three-mirror design or TMA telescope.

Investigations of the plane-aspheric mirror have been discussed to define the best shape freeform surface of this non-centred system. The freeform surface is with elliptical symmetry and provides balance of the quadratic terms with respect to the bi-quadratic terms.

Temporarily avoiding the folding-flat mirror, let assume a two-mirror non-centred system where the primary mirror M1 is a freeform and the secondary mirror M2 is a concave spherical surface of radius of curvature *R*. The input beams are circular collimated beams merging at various field angles of a telescope and define along the M1 freeform mirror an elliptical pupil due to inclination angle *i*. A convenient value of the inclination angle allows the M2 mirror to avoid any obstruction and provide

focusing very closely to mid-distance between M1 and M2, then very near the distance $R/2$ from M1 (Figure 2).

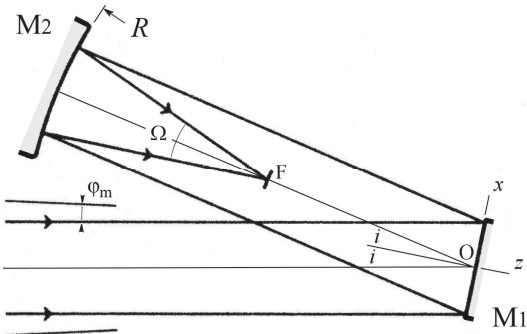

**Figure 2.** Schematic of a reflective two-mirror Schmidt telescope. The centre of curvature of the spherical mirror M2 is located at the vertex of the plane-aspheric freeform mirror M1. The incident beams have circular cross-sections with input pupil located at mirror M1. Deviation $2i$ occurs at the principal rays. The semi-field maximum angle—here assumed circular—is denoted $\varphi_{\mathrm{m}}$.

Let us denote $x_{\mathrm{m}}$ and $y_{\mathrm{m}}$ the semi-axes of the elliptic clear-aperture on M1 primary mirror. If the $y$-axis is perpendicular to the symmetry plane $x, z$ of the two-mirror telescope, one defines

$$\Omega = R/4y_{\mathrm{m}} \tag{1}$$

a dimensionless ratio $\Omega$, where $y_{\mathrm{m}}$ is the semi-clear-aperture of the beams in $y$-direction, that is half-pupil size in y-direction. The *focal-ratio* of the two-mirror telescope is denoted $f/\Omega$.

It is the interesting to investigate the performance in the field of view of a 100% obstructed two-mirror telescope which then is a centred system. Comparing the blur images, the free parameter is the ratio that characterize the meridian profile section of the primary mirror M1. This ratio, or aspect ratio, can be expressed by $k = r_O{}^2/r_m{}^2$ where $r_O$ is the radius of null-power zone and $r_m$ that of clear aperture. The blur residual sizes as a function of parameter $k$ are displayed by Figure 3 [4,6].

In order to avoid 100% beam obstruction the primary mirror M1 must be tilted at a convenient inclination angle $i$. It can be shown that the shape $Z_{\mathrm{Opt}}$ of the M1 freeform mirror is an anamorphic shape and expressed in first approximation by [4]

$$Z_{Opt} \simeq \frac{s}{\cos i} \left[ \frac{3}{2^7 \Omega^2 R} (h^2 x^2 + y^2) - \frac{1}{8R^3} (h^2 x^2 + y^2)^2 \right], \qquad with\ h^2 = \cos^2 i, \tag{2}$$

where dimensionless coefficients could be considered as an under-correction parameter slightly smaller than unity ($0.990 < s < 1$). In fact, for a tilt angle of M1—that is a non-centred system—coefficients does not operates and must just be set to $s = 1$. A preliminary analysis of the two-mirror system shows that for four direction points of a circular field of view, the largest blur image occurs at the largest deviation angle of the field along the $x$-axis. In $y$-direction sideways blur images have an average size.

The length from the vertex O of the M1 mirror to the focus F can be derived as [4]

$$OF = \frac{1}{2} \left( 1 + \frac{3}{2^6 \Omega^2} \right) R \tag{3}$$

showing that this distance is slightly larger than the Gaussian distance $R/2$. The optimal size of blur images for an $f/4$ two-mirror anastigmat telescope over the field of view are displayed by Figure 4.

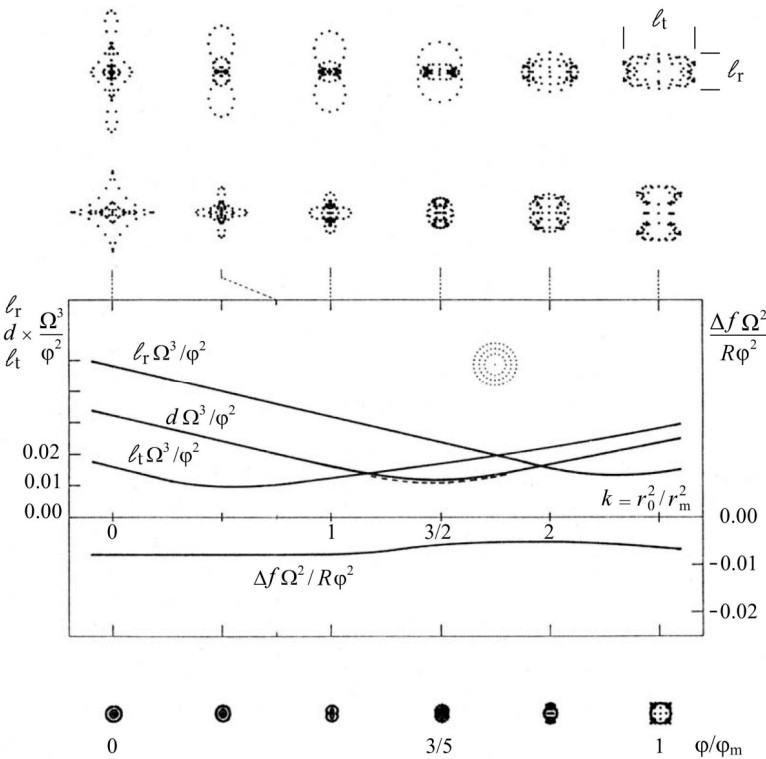

**Figure 3.** Off-axis 5th-order aberration residuals of a two-mirror centred-system reflective Schmidt—100% obstruction—as a function of the $k$-ratio, $k = r_O^2/r_m^2$, that defines location of the null-power optical zone. Plotted parameters are: radius of curvature $R$ of M2 spherical concave mirror, $\Omega$ of the aperture-ratio $f/\Omega$, angle of semi-field of view $\varphi$, radial and tangential blur sizes $\ell_r$, $\ell_t$ and diameter of best blur size $d$. [*1st line*] Spot diagram of blur images on a sphere centred at the centre of curvature of M1 mirror that provides stigmatism on-axis. [*2nd line*] Spot diagram of blur images on a sphere centred at the centre of curvature of M1 mirror that provides the best images. [*Bottom line*] Spots with $k = 3/2$ for field angles $\varphi/\varphi_m$ varying from 0 to 1, obtained by using a slight under-correction factor of M1 mirror $s = \cos\varphi_m$ and defocus $\Delta f$. The three rows of spot images are at same scale. Lemaitre showed that [6] the final diameter of residual blurs—or *angular resolution of reflective Schmidt*—is then $d_{\text{Resol}} = 3 \, \varphi_m^2/256\Omega^3$.

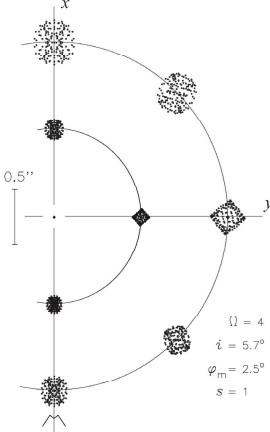

**Figure 4.** Best residual aberrations of an $f/4$ two-mirror reflective Schmidt ($\Omega = 4$) in the *non-centred system* form. Inclination angle $i = 5.7°$. Semi-field of view $\varphi_m = 2.5°$. The under-correction parameter is just set to $s = 1$. The radius of curvature of the spherical focal surface is also given by $R_{\text{FoV}} = \text{OF}$ in Equation (3). The largest blur image corresponds to that with highest deviation of the FoV.

Preliminarily analyses also show that the M1 mirror shape has opposite signs between quadratic and bi- quadratic terms. The shape is given by Equation (2) and represented by Figure 5 in either directions $x$ or $y$ in dimensionless coordinates of $\rho$.

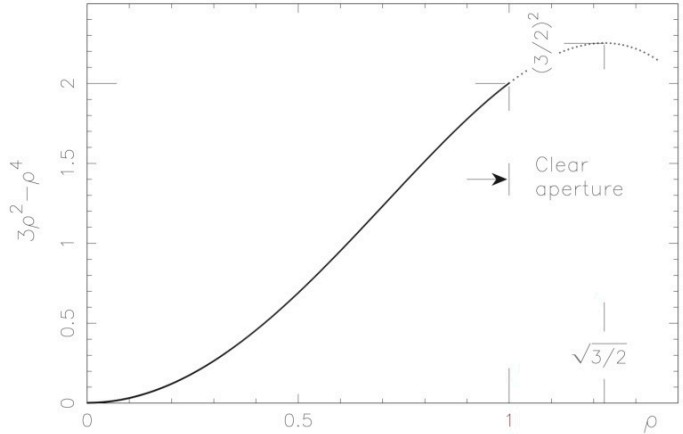

**Figure 5.** The freeform primary mirror, of biaxial symmetry, is generated by homothetic ellipses having principal lengths in a cos*i* ratio. One shows that whatever $x$ or $y$ directions the null-power zone is outside the M1 clear-aperture and in a geometrical ratio $\rho_0 = \sqrt{3/2}\,\rho_{max} \simeq 1.224\rho_{max}$ since $k = 3/2$.

Dimensionless coordinates $z$, $\rho$ have been normalized for a clear semi-aperture $\rho = 1$, presently in the $y$-direction, with a maximum sag

$$z(1) = 3\rho^2 - \rho^4 = 2, \tag{4}$$

which leads to algebraically opposite curvatures $d^2z/d\rho^2$ for $\rho = 0$ and $\rho = 1$.

The conclusions from the best optical design of an all-reflective two-mirror Schmidt telescope with *optimal angular resolution* are the followings [4,6]:

1. *For a circular incident beam, the elliptical clear-aperture of the primary mirror is $\sqrt{3/2}$ times smaller to that of the null-power zone ellipse.*

2. *The angular resolution $d_{\mathrm{NC}}$ of a reflective non-centred two-mirror telescope is*

$$d_{NC} = \frac{3}{256\Omega^3}\left(\frac{3}{2}i + \varphi_m\right)\varphi_m \tag{5}$$

3. *The primary mirror of a non-centred two-mirror system provides the algebraic balance of second derivative extremals. Therefore, its central curvature is opposite to the local curvatures at edge.*

## 3. Elasticity Design and Deformable Primary-Mirror Substrate

Active optics preliminarily analysis of a Schmidt-type primary mirror M1 leads to investigate deformable substrates and use of stress mirror polishing (SMP). We consider hereafter the thin plate theory of plates with development to elliptical plates.

Let consider the system coordinates of an elliptical plate and denote $n$ the normal to the contour $C$ of the surface. Equation of $C$ is represented by (Figure 6)

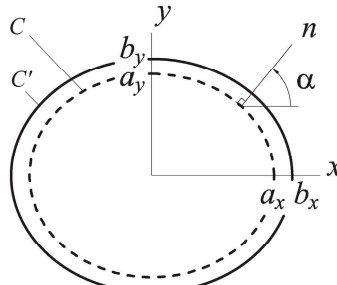

**Figure 6.** Top view of an *elliptical vase form*. The clear aperture of M1 mirror is included into a smaller surface to that of a constant thickness plate delimited by elliptical contour *C* (*dotted line*) and defined by semi axe radii ($a_x$, $a_y$). An outer ring is built-in to the plate at contour *C* where *n*-directions are normal to *C*. The outer ring is delimited by a homothetic elliptic contour *C'* defined by semi-axes radii ($b_x$, $b_y$).

The bi-Laplacian equation of the flexure where a uniform load *q* is applied to the inner plate is [4,7]

$$\nabla^4 z \equiv \partial^4 z/\partial x^4 + 2\partial^4 z/\partial x^2 \partial y^2 + \partial^4 z/\partial y^4 = q/D, \tag{6}$$

where the rigidity *D* is [4,7]

$$D = Et^3/[12(1 - \nu^2)], \tag{7}$$

and *E* the Young's modulus, $\nu$ the Poisson ratio and *t* the constant thickness of the plate over *C*.

The equation of homothetic ellipses $C(a_x, a_y)$ or $C'(b_x, b_y)$ can be expressed by quadratic forms. Ellipse curve *C* writes

$$\frac{x^2}{a_x^2} + \frac{y^2}{a_y^2} - 1 = 0. \tag{8}$$

Remaining within the optics theory of third-order aberrations, such optical freeform surfaces can be obtained from elastic bending by mean of the following conditions:

- a flat and constant-thickness plate, *t* = *constant*,
- a uniform load *q* applied to inner surface of substrate,
- and a link at the edge to an elliptic contour expressed by a *built-in edge*, or a *clamped edge*, that is where the slope is null all along the contour *C* of the plate.

Assuming that the three conditions below are satisfied, the analytic theory of thin plates allows deriving a biquadratic flexure $Z_{Elas}(x,y)$ in the form [4,7],

$$Z_{Elas} = z_0 \left(1 - \frac{x^2}{a_x^2} - \frac{y^2}{a_y^2}\right)^2, \tag{9}$$

where $z_0$ is the sag at origin and where $a_x$ and $a_y$ are semi-axes corresponding to the elliptic null-power zone of the principal directions of contour. The flexural sag $z_0$ is obtained from substitution of Equation (9) into biharmonic Equation (6). This leads to

$$z_0 = \frac{q}{8D} \frac{a_x^4 a_y^4}{3a_x^4 + 2a_x^2 a_y^2 + 3a_y^4}. \tag{10}$$

The null-power zone contour *C* is larger by a factor $\sqrt{3/2}$ to that of the clear-aperture (cf. Figure 3).

The dimensions of semi-axes $a_x$ and $a_y$ at *C* of the built-in ellipse – or null-power zone –relatively to that of clear-semi-apertures $x_m$ and $y_m$ are

$$a_x^2 = 3x_m^2/2 = 3y_m^2/2\cos^2 i \quad and \quad a_y^2 = 3y_m^2/2. \tag{11}$$

From Equations (2) and (1), in setting $s = 1$ and $x = 0$ and for the built-in radius $y = \sqrt{3/2}\, y_m$ of the vase form, we obtain the amplitude of the flexure $z_0$ in Equation (9), as

$$z_0 = \frac{9 y_m}{2^{11} \Omega^3 \cos i}. \tag{12}$$

From Equations (10)–(12) we obtain, after simplification, the thickness $t$ of the inner plate

$$t = 8\Omega \left[ \frac{2(1 - v^2 \cos i)}{3(3 + 2\cos^2 i + 3\cos^4 i)} \frac{q}{E} \right]^{1/3} y_m. \tag{13}$$

This defines the execution conditions and elasticity parameters of an elliptical plate where the clear aperture of primary mirror M1 uses a somewhat smaller area than the total built-in surface as delimited by ellipse $C$.

The conclusions from a best optical design of the primary mirror M1 of a two-mirror Schmidt telescope, as corresponding to the profile displayed by Figure 3, are as follow [4]:

1.  *A built-in elliptic vase-form is useful to obtain easily the primary mirror of a two-mirror anastigmat.*
2.  *The elliptic null-power zone at the plate built-in contour is $\sqrt{3/2}$ times larger than that of the elliptic clear-aperture of the primary mirror.*
3.  *The total sag of the built-in contour is 9/8 times larger than that of its optical clear-aperture.*

The optimization of a built-in substrate, of course, requires losing some of the outer surface which then is not usable by an amount of 33% outside the clear-aperture area. It is clear that a flat deformable M1 substrate, conjugated with an elliptic inner contour, provides most interesting advantages in practice with a *built-in condition* (Figure 7).

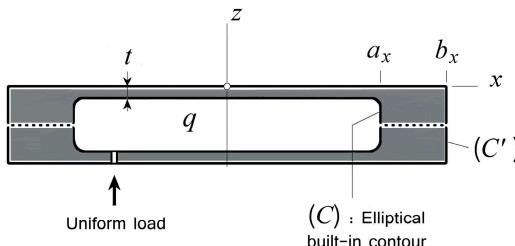

**Figure 7.** Section of an *elliptical closed vase form*. This form is made two vase-forms oppositely jointed together in a *built-in* link. The uniform load $q$ is applied all inside the deformable substrate over the elliptical contour C of semi-axe radii $(a_x, a_y)$. Outer contour $C'$ can be made circular if $(b_x, b_y)$ are sufficiently larger than semi-axe radii of C.

An extremely rigid outer ring designed for a *closed vase form*, while polished flat at rest, allows the primary mirror to generate by uniform loading a bi-symmetric optical surface made of *homothetic ellipses* (Figure 8).

The case of a perfect *built-in condition* was applied to the design and construction of several aspherized grating spectrographs (4) such as UVPF of CFHT, MARLYs of OHP and PMO, CARELEC of OHP, OSIRIS of ODIN space mission. It was recently applied to the design and construction of aspherized gratings for the FIREBall balloon experiment [8] using a 1-meter telescope and multi-object spectrograph. The processing was as follows: (i) A plane reflective diffraction grating was first deposited on a deformable stainless steel matrix without stressing; (ii) the freeform bi-quadratic surface of this deformable matrix was bent by inner air pressure loading; (iii) the freeform gratings were obtained from resin replication during controlled stressing onto Zerodur vitro ceramic rigid substrate; and (iv) final aspheric shape of the gratings was obtained by elastic relaxation [4] (Figure 9).

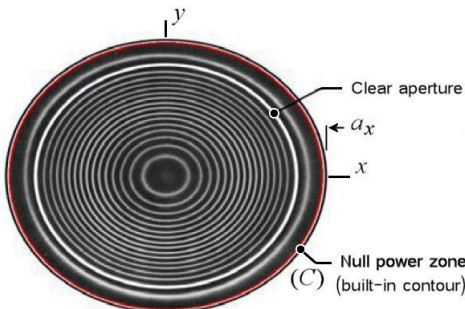

**Figure 8.** Iso-level lines of a primary mirror M1 generating homothetical ellipses from perfect built-in condition at contour C. Elliptic dimensions of the null-power zone over those of clear-aperture must be set in a ratio $\sqrt{3/2}$. The algebraic balance of meridian curvatures is then achieved at centre and at clear-aperture.

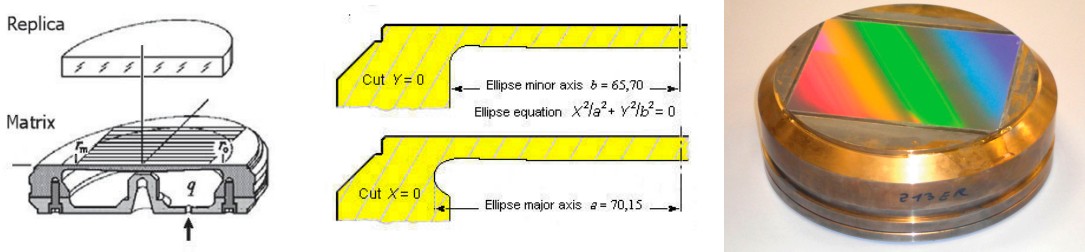

**Figure 9.** [*Left*] Active optics aspherization of gratings achieved by double replication technique of a metal deformable matrix. A *quasi-constant thickness* active zone is *clamped*—or *built-in*—to a rigid outer ring and is closed backside for air pressure load control. [*Centre*]. For FIREBall, the whole deformable matrix is an axisymmetric piece expect for its inner built-in contour which is *elliptical*. [*Right*] Deformable matrix with the grating before replication (LAM).

Compared to the case of or a deformable M1 mirror—or a deformable matrix—where a perfect *built-in* condition leads to some outside loosen optical area of the freeform (a clear aperture somewhat smaller to that of the built-in zero power zone), we investigate hereafter a *closed-form substrate* with a *radially thinned outer elliptic cylinder*. The optimized freeform surface for MESSIER proposal will then provide almost the telescope theoretical angular resolution by use of a *semi-built-in edge condition* of the mirror substrate.

## 4. A TMA Telescope Proposal for MESSIER

Our proposal of MESSIER experiment is named in honour to French astronomer Charles Messier who started compiling in 1774 his famous Messier Catalogue of diffuse non-cometary objects. For instance the Crab Nebulae, named object "M1," is a bright supernova remain recorded by Chinese astronomer in 1054. The present MESSIER proposal is dedicated to the detection of extremely low surface brightness objects. A detailed description of the science objectives and instrument design for this space mission can be found in [9].

A three mirror anastigmat (TMA) telescope proposal should provide detection of extremely low surface brightness. MESSIER science goals require that any optical design should deliver a detection as a detection as low as 32 magnitude/arcsec$^2$ in the optical range. A space-based telescope option should provide 37 magnitude/arcsec$^2$ in UV at 200 nm. The preliminary ground-based telescope here investigated as a prototype should be useful for spectral ranges in blue, red and infrared regions. The main features require a particular optical design as follow:

1. a wide field three mirror anastigmat telescope with fast f-ratio,
2. a distortion-free field of view at least in one direction,
3. a curved-field detector for a natural curved reflective Schmidt,

4.    a time delay integration by use of drift-scan techniques,
5.    no spider in the beams can be placed in the optical train.

The final space-based proposal, restrained to extremely low brightness objects in ultraviolet imaging, would be a 50-cm aperture telescope. For the optical range, our preliminary plan for MESSIER is to develop and build a 30-cm aperture ground-based telescope fulfilling all above features of the optical design. Preliminary proposed designs can be found in [5,10].

### 4.1. Optical Design and Ray Tracing Modelling

Before studying a space model, we propose development of a ground-based prototype telescope is a three-mirror anastigmat (TMA) with an f-ratio at $f/2.5$.

One may notice a classical design as a two-element anastigmat design where the first optical element is a refractive aspherical plate. Minimization of field aberrations is achieved by a *balance of the slopes—that is balance of 1st-order derivative*—where sphero-chromatism variations are dominating aberration residuals.

Now using a mirror—instead of a refractive plate—as a first element M1 of a two-mirror anastigmat, the best angular resolution over the field of view is achieved by a *balance of meridian curvatures—that is balance of the 2nd-order derivative*—of the aspherical mirror or diffraction grating [4].

The proposed design is made of freeform primary mirror M1, followed by holed flat secondary mirror M2 and then a spherical concave tertiary mirror M3. The centre of curvature of M3 is located at the vertex of M1which is a basic configuration for a reflective Schmidt concept. The unfolded version, with only two mirrors, do not alter anastigmatic image quality (Figure 13). The optics parameters are optimized for a convex field of view (Table 1). The three-mirror system gives unobstructed access to detector and avoid any spider in the field of view (Figures 10 and 11).

**Table 1.** MESSIER optical design parameters.

| | |
|---|---|
| Telescope angular resolution | 2 arcsec |
| On-axis circular beam entrance | 256 mm |
| Focal length $f'$ | 890 mm |
| Focal-ratio f/Ω | f/2.5 |
| Deviation angle $2i$ | 22° |
| M1 Elliptic clear aperture $2x_m \times 2y_m$ | $356 \times 362.7$ |
| Angular FOV $2\varphi mx \times 2\varphi my$ | $1.6° \times 2.6°$ |
| Linear FOV | $25 \times 40$ mm$^2$ |
| M3 mirror curvature radius $R3$ | $-1769$ mm |
| UBK7 filters & SiO$_2$ cryostat—Thicknesses | 2 & 5 mm |
| Detector field curvature radius $R_{FO}$ | $-890$ mm |

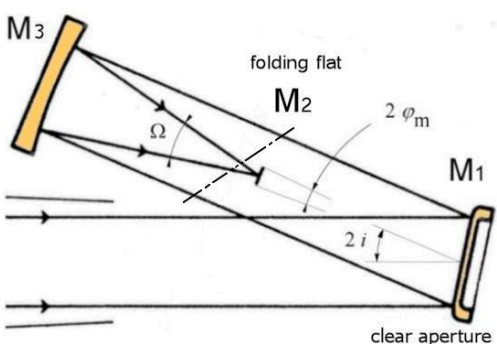

**Figure 10.** Schematic of Messier reflective Schmidt (M2 holed flat not shown).

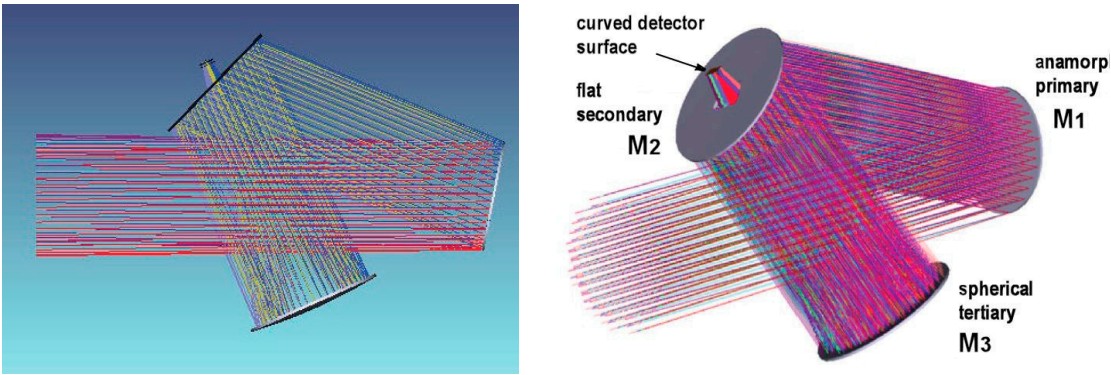

**Figure 11.** [*Left*] MESSIER layout reflective TMA in the symmetry plane. [*Right*] 3-D view with detector (LAM).

The primary mirror surface M1 can be defined by the aspheric anamorphic equation as derived from Zemax optics ray-tracing code—which is somewhat similar to Equaiton (2)—and denoted

$$Z_{Opt} = \frac{C_x x^2 + C_y y^2}{1 + \sqrt{1 - (1 + K_x)C_x^2 x^2 - (1 + K_y)C_y^2 y^2}} + AR[(1 - AP)x^2 + (1 + AP)y^2]^2, \quad (14)$$

where now in all this section with Zemax code $(y, z)$ is the symmetry plane of the telescope.

Restraining to the case of a simply bi-curvature surface for the first quadratic term, then setting $K_x = K_y = -1$, denominator reduces to unity. The result from Zemax modelling optimization provides the four coefficients, $C_x = -3.598 \times 10^{-6}$ mm$^{-1}$, $C_y = -3.467 \times 10^{-6}$ mm$^{-1}$, $AR = 2.203 \times 10^{-11}$ mm$^{-3}$ and $AP = -0.01854$.

The *total sag of clear aperture* in $x$-direction (i.e., off-symmetry plane) for $x_{\max} = 178$ mm is then $Z_{\mathrm{Opt-max}} = -35.70$ µm. *Zemax* iterations lead to RMS residual blur images in agreement with predicted angular resolution (Figure 12).

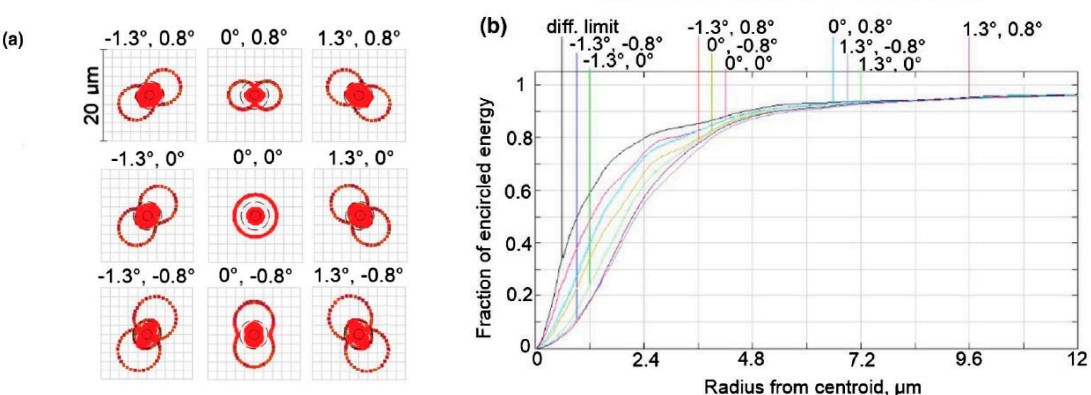

**Figure 12.** Telescope image quality—curved FOV. (**a**) Spot diagram. (**b**) Diffraction encircled energy.

Results from Zemax modelling and spot diagram one shows that the maximum RMS blur residuals is $\phi_{\mathrm{Zemax}} = 5.5$ µm or $d_{\mathrm{NC}} = 1.27$ arcsec. From Equation (5) and a same diagonal semi-FOV (1.52°) and same parameters in Table 2, one also obtains for MESIER proposal the theoretical angular resolution [4,6],

$$d_{NC} = \frac{3}{256\Omega^3}\left(\frac{3}{2}i + \varphi_m\right)\varphi_m = 1.29 \text{ arcsec}, \quad (15)$$

showing that results from modelled and theoretical angular resolutions are similar.

One of the key requirements was to provide a free distortion design. Over a diagonal field of view of maximum radius $\varphi_m = 1.5°$ it was shown that for a curved FoV distortion $\Delta\varphi/\varphi_m$ remains smaller than $2 \times 10^{-4}$ and thus is negligible.

The telescope image quality is estimated through its spot sizes in the diagram. Diagram indicates an image quality convenient for seeing limited conditions and close to the diffraction limit. Contribution of residual aberrations is 1.3 arcsec RMS over the field of view.

Residual spot radius values take into account a UBK7 colour filter 2 mm thick where a spectral band can be selected in a filter wheel with 5 wavebands. An optional 5 mm thickness $SiO_2$ cryostat window is planned for optical design of the ground-based prototype telescope (Table 2).

**Table 2.** Image quality of the telescope with use of filters.

| Waveband [nm] | Spot RMS Radius in the FoV Centre [μm] | Spot RMS Radius at the FoV Edge [μm] |
| --- | --- | --- |
| 350–410 | 1.8 | 3.7 |
| 400–560 | 2.3 | 3.8 |
| 550–700 | 1.7 | 3.4 |
| 680–860 | 1.7 | 3.3 |
| 860–980 | 1.7 | 3.3 |

### 4.2. Elasticity Modelling of a Freeform Primary Mirror

The requested aspheric surface with non-rotational symmetry of the primary mirror surface refers to an optical surface also called a *freeform surface*. The present freeform surface for our MESSIER primary mirror is made of *homothetic-ellipse iso-level lines*. This surface is to be designed through active optics methods where the deformable substrate is aspherized by a plane surfacing under stress—an active optics method also called stress mirror polishing (SFP). The principle uses a uniform load applied and controlled inside a *closed vase form* whilst a plane polishing is operated with a full-size tool. The final process delivers the required shape after elastic relaxation of the load [5,6,10,11].

The *closed vase form*—or *closed biplate*—is made of facing twin elliptical vase form blanks in Zerodur assembled together at the end of their outer rings through a layer of 100–150 μm thickness 3M DP490 Epoxy, where elasticity constants are Poisson's ratio $\nu = 0.38$ and Young's modulus $E = 659$ MPa [12]. Elasticity constants of the blanks in Zerodur are Poisson's ratio $\nu = 0.243$ and Young's modulus $E = 90.2$ GPa (Figure 13).

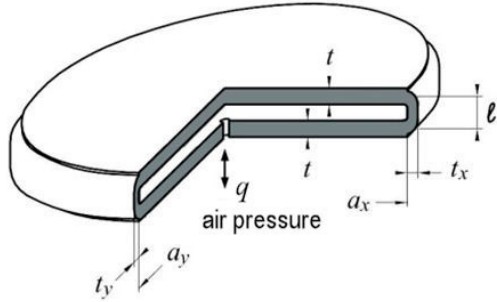

**Figure 13.** Elasticity design of primary mirror substrate as a *closed vase form* or *closed biplate* made of two identical vase vitro-ceramic material linked together with epoxy. The radial thicknesses $(t_x, t_y)$ and height $\ell$ of the outer cylinder provides a semi-built-in boundary which somewhat reduces the size of inner ring radii $(a_x, a_y)$ with respect to that of clear aperture $(x_m, y_m)$. NB.: From anamorphose $x_m/y_m = a_x/a_y = t_x/t_y = \cos i$.

Modelling with Nastran code led us to make cross optimizations with Zemax code. The intensity of uniform loading $q$ and final geometry of the *closed vase form* mirror substrate are displayed by Table 3.

The final finite element modelling by Pascal Vola display the displacements (Figure 14) and stresses (Figure 15).

**Table 3.** Uniform load and geometry of *closed vase form* deformable substrate of MESSIER primary mirror. Substrate in Zerodur vitro-ceramic. Mirror aspherization is achieved after elastic relaxation from SMP method and full-size tool plane polishing.

| | |
|---|---|
| Uniform load of constant pressure | $q = 0.687 \times 10^5$ Pa |
| Axial thickness of parallel plates | $t = 18$ mm each |
| Inner axial separation of closed plates | 20 mm |
| Outer thickness of closed form | 56 mm |
| Inner diameters of elliptic cylinder | $2a_x \times 2a_y = 356 \times 362.7$ mm |
| Cylinder radial thicknesses | $t_x, t_y = 18$ and $18.33$ mm |
| Distance between middle surface of plates | $\ell = 38$ mm |
| Inner and outer round-corner radii | $R_C = 8$ mm and 16 mm |

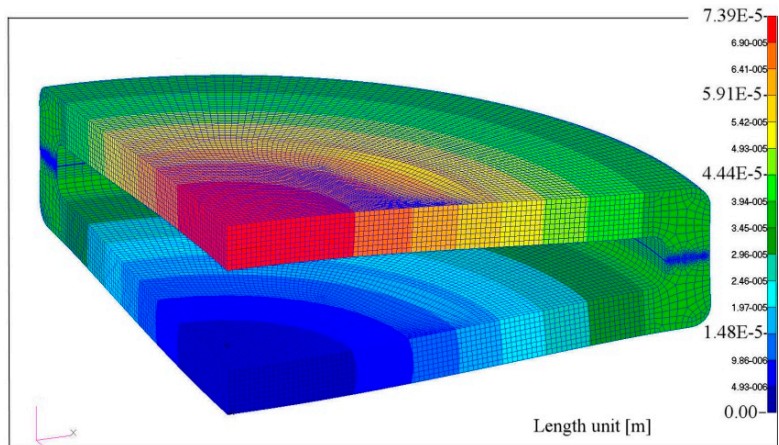

**Figure 14.** Displacements of M1 primary mirror substrate as a *closed vase form* with FEA Nastran code. All elements are hexahedra. Boundaries are expressed at the origin of displacements $x = y = z = 0$ and freedom along three radial directions in plane *x-y* both at back surface of the figure. Total axial displacement 69.2 μm (LAM).

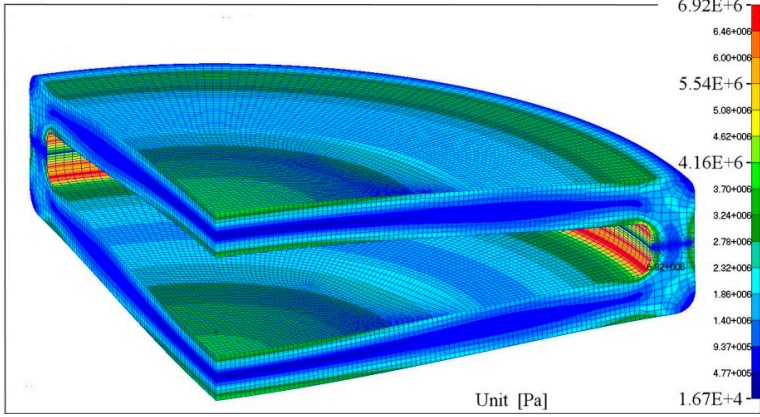

**Figure 15.** Stress distribution of the *closed vase form* during stress polishing. The maximum tensile stress arises along the internal round corners of the elliptical rings with $\sigma_{max} = 6.92$ MPa. For Schott-Zerodur this value is much smaller than the ultimate strength $\sigma = 51$ MPa for a 1-month loading time duration [4]. One may notice that at the symmetry plane epoxy link reduces to $\sigma \simeq 1$ MPa (LAM).

The general law of algebraically *balancing the second-order* derivatives applies on principal directions of a *pupil mirror*—as stated by Equation (4)—for a circular or elliptical clear aperture. In dimensionless coordinates and any $x$ or $y$ orthogonal directions, if the clear aperture radii is denoted $\rho_\mathrm{m}$, then we must obtain algebraically opposite local curvature values for $\rho = 0$ and $\rho = \rho_\mathrm{m}$.

One has shown [4] that with a perfect built-in condition – that is for a closed vase form with moderate ellipticity and a ring of large radial thickness, say, at least five times larger than the plate axial thickness—the flexure provides an elliptic *null-power zone* radius $\rho_0$ where the size of the *clear aperture* radius $\rho_\mathrm{m}$ is in the ratio $\rho_0/\rho_\mathrm{m} = \sqrt{3/2} \simeq 1.224$. This means that the useful optical area is convenient for $\rho \in [\rho_\mathrm{m}, \rho_0]$ but unacceptable for $\rho \in [\rho_\mathrm{m}, \rho_0]$.

From our cross optimizations with Zemax and Nastran codes the ring of the closed vase form provides a noticeable decrease in flexural rigidity, which is equivalent to a *semi-built-in condition*. Then, compared to a perfect built-in assembly and from our results described in latter subsection, the radii ratio between null-power zone $\rho_0$ and clear aperture $\rho_\mathrm{m}$ has become after optimized modelling

$$\rho_0/\rho_\mathrm{m} = 1.118 \tag{16}$$

This ratio is *significantly* smaller than 1.224. Then from Zemax ray tracing optimizations, we now take benefit of an important result: the angular resolution is not substantially modified in using the optical surface also up to the inner zone of the ring. The flat deformable surface of the closed vase form requires use of stress polishing by plane super-polishing, which then avoids any ripple errors of the freeform surface.

Closed vase form Messier primary mirror can be aspherized up to its inner part of the semi-built-in elliptic ring without any lost in optical area and angular resolution.

### 4.3. Optical Testing of MESSIER Freeform Primary Mirror

Optical testing of the freeform primary mirror must be a precise measurement because of its anamorphic shape. From Equation (14) the clear aperture shape presents a total sag of $Z_{\mathrm{Opt-max}} = -35.70$ μm for $x_{\max} = 178$ mm in $x$-direction, that is in the off-symmetry telescope plane. Several optical tests could be considered mainly based on a null-test system [6]. For instance this involves lens compensators [13] and computed-generated holograms [14] and their combinations.

We adopted a singlet lens compensator as component already existing at the lab and providing exactly the correct compensation level of the rotational symmetry mode, that is, 3rd-order spherical aberration compensation of M1 primary mirror. This lens, also called Fizeau or Marioge lens, is plano-convex made in Zerodur-Schott with following parameters, axial thickness $t_\mathrm{L} = 62$ mm, $R_{1\mathrm{L}} = \infty$, $R_{2\mathrm{L}} = 1180$ mm, $D_\mathrm{L} = 380$ mm, used on elliptical clear aperture $356 \times 362.7$ mm. The axial separation to primary mirror is 10 mm. Remaining aberration is then a balanced anamorphose term to be accurately calibrated by He-Ne interferometry (Figure 16).

Another important feature of MESSIER telescope proposal is the selection of a *curved detector*, $R_{\mathrm{FOV}} \simeq -f = -R_3/2$ (cf. Table 1), which allows a *distortion free* design over an active area of $40 \times 25$ mm$^2$. This technology is presently under development by use of either *variable curvature mirrors* (VCMs) or *toroid deformed mirrors* [4,5]. References can be found on curved detectors in Muslimov paper [15].

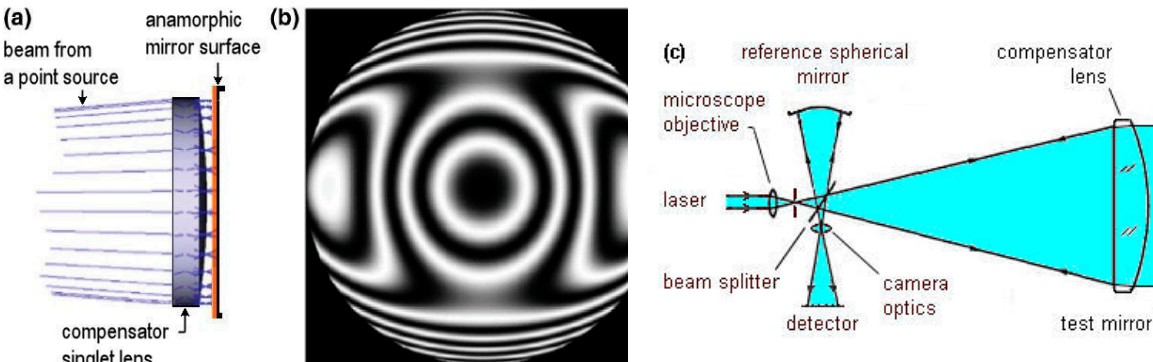

**Figure 16.** Null-test singlet-lens aberration compensator for MESSIER freeform primary mirror. (**a**) Scheme mounting of the plano-convex lens that compensate for spherical aberration of the primary mirror surface. (**b**) Simulated He-Ne interferogram of remaining non-axisymmetric fringes to be calibrated. (**c**) Interferences can be obtained from a two-arm interferometer (LAM).

## 5. Conclusions

Modelling of freeform surfaces by development of *active optics techniques* or *stress mirror polishing* provides extremely smooth surfaces. Generated from elastically relaxed plane or spherical polishing with *full-size aperture tools* these surfaces are free from ripple errors.

Applied to MESSIER TMA telescope proposal the best angular resolution of a *non-centred optical designs* corresponds to a primary mirror freeform optical surface made of homothetic elliptical isolevel lines. It has been shown that this leads to *algebraic balance of local curvatures* – that is *balance of second derivatives* of the optical surface

Beside the facility to obtain easily a freeform aspheric from elastic relaxation of a *closed vase form* primary mirror, MESSIER proposal would also benefit from free distortion because of its *naturally* curved FoV. This field curvature shall be associated to a *curved detector*.

**Author Contributions:** E.M. investigated the telescope optical design with Zemax raytracing optimization and developed a useful optical quasi-null-test optimization for the freeform mirror. P.V. developed an accurate finite element analysis of the closed vase form mirror with Nastran code for stress polishing and elastic relaxation technique. G.R.L. established the law that applies for the optimal shape to be given freeform mirrors and freeform gratings of reflective Schmidt systems, developed a raytracing code with Seidel modes freeform surfaces, investigated the telescope optical design with Zemax code anamorphic surface optimizations, and introduced the concept of closed vase form mirror.

**Conflicts of Interest:** The authors declare no conflict of interest.

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
