# Peer review of "Active Optics in Astronomy: Freeform Mirror for the MESSIER Telescope Proposal"

_mca, doi:10.3390/mca24010002_

Round 1

Reviewer 1 Report

The first author of this paper, Gerard Lemaitre, has originally invented and developed several active optics methods since 1970’s at the Laboratoire d’Optique de l’Observatoire de Marseille (LOOM) in the following way:

The mirror is polished spherical in an unstressed state, then aspherised by permanent stress (normal active optics in operation);

The mirror is polished spherical under stress, then assumes the required aspheric shape after stress release (normal stress polishing);

The mirror is submitted to a combination of both the 1) and 2) methods.

These methods present the fundamental advantage of generating high aspherics and freeform optics as smooth as those obtained when figuring of spheres and planes.

It seems to me that this research paper of ‘‘Active Optics in Astronomy-Freeform Mirror for the MESSIER Telescope Proposal’’ presents a novel design of the wide-field low-central-obscuration three-mirror-anastigmat telescope, including to use a freeform primary mirror obtained from active optics with closed-vase form, for MESSIER space mission.  This is an all reflective Schmidt design with a holed flat M2 for beam folding, in order to minimize the central obscuration in the telescope. The novel active optics method, as proposed by the authors in the paper, allows them to feasibly obtain a freeform mirror from elastic relaxation of a closed vase form primary mirror. In order to meet the mission requirements of MESSIER proposal, a curved (3D) sensor (focal plane array) will be used on the focal plane of the telescope for free distortion and optical resolution required over the fields, due to the residual field curvature in the telescope. With this submitted version, my main comments about this research paper are as follows:

1. This is a very interesting research paper with a good design and analysis. Especially, the authors have developed an active-optics technique, or called stress mirror polishing, to obtain a freeform mirror from elastic relaxation with a closed vase form primary mirror.

2. All the optical calculation and analysis are correct as shown in the Section-2 of the paper, and the elasticity design and analysis with the freeform primary mirror substrate, as given in Section-3, are reasonable and relevant to the telescope design they proposed.

3. The optical design with the telescope by ray tracing with Zemax is correct and has a good agreement with their previous analysis, as given in Section-2 and Section-4.

4.  In Section-4, they have also given an elasticity analysis for the closed-vase primary mirror, by using FEA Natstran opto-mechanical code, with a reasonable deformation as required for the mirror’s shape in the telescope.

5. For some freeform surfaces it is true, especially when considering the optical testing with the surfaces to be necessary before the system assembly and integration. Often, the most significant impediment to progress is the optical testing of these freeform-surfaces. The preferred method for imaging optics is always interferometry, which can perfectly limit the maximum slope that a surface may contain. The optical testing of MESSIER freeform primary mirror, proposed by the authors in Section-4.3, is an interesting setup and method for the freeform tests and verification after mirror’s fabrication.

6. On page-9, there are some typing errors in Table 2, and it seems to me that they should be like such as:

On-axis circular beam entrance diameter            356 mm

Focal length f                                                       890 mm

Focal-ratio                                                            f/2.5

Deviation angle 2i                                                22 degrees

In my opinion: This is a new idea of the freeform design and fabrication for space optics application. My recommendation is to accept this research paper as a novel optical design and fabrication for freeform optics in space application, after making a minor correction of typing errors in Table 2 (on page-9) as well as a bit of English error on page-4.

Author Response

ANSWER to Reviewer-1 AT THE END – in blue,

and, My own comment to Editor AFTER THE END – in blue

Comments and Suggestions for Authors

The first author of this paper, Gerard Lemaitre, has originally invented and developed several active optics methods since 1970’s at the Laboratoire d’Optique de l’Observatoire de Marseille (LOOM) in the following way :

The mirror is polished spherical in an unstressed state, then aspherised by permanent stress (normal active optics in operation);

The mirror is polished spherical under stress, then assumes the required aspheric shape after stress release (normal stress polishing);

The mirror is submitted to a combination of both the 1) and 2) methods.

These methods present the fundamental advantage of generating high aspherics and freeform optics as smooth as those obtained when figuring of spheres and planes.

It seems to me that this research paper of ‘‘Active Optics in Astronomy-Freeform Mirror for the MESSIER Telescope Proposal’’ presents a novel design of the wide-field low-central-obscuration three-mirror-anastigmat telescope, including to use a freeform primary mirror obtained from active optics with closed-vase form, for MESSIER space mission.  This is an all reflective Schmidt design with a holed flat M2 for beam folding, in order to minimize the central obscuration in the telescope. The novel active optics method, as proposed by the authors in the paper, allows them to feasibly obtain a freeform mirror from elastic relaxation of a closed vase form primary mirror. In order to meet the mission requirements of MESSIER proposal, a curved (3D) sensor (focal plane array) will be used on the focal plane of the telescope for free distortion and optical resolution required over the fields, due to the residual field curvature in the telescope. With this submitted version, my main comments about this research paper are as follows:

1. This is a very interesting research paper with a good design and analysis. Especially, the authors have developed an active-optics technique, or called stress mirror polishing, to obtain a freeform mirror from elastic relaxation with a closed vase form primary mirror.

2. All the optical calculation and analysis are correct as shown in the Section-2 of the paper, and the elasticity design and analysis with the freeform primary mirror substrate, as given in Section-3, are reasonable and relevant to the telescope design they proposed.

3. The optical design with the telescope by ray tracing with Zemax is correct and has a good agreement with their previous analysis, as given in Section-2 and Section-4.

4.  In Section-4, they have also given an elasticity analysis for the closed-vase primary mirror, by using FEA Natstran opto-mechanical code, with a reasonable deformation as required for the mirror’s shape in the telescope.

5. For some freeform surfaces it is true, especially when considering the optical testing with the surfaces to be necessary before the system assembly and integration. Often, the most significant impediment to progress is the optical testing of these freeform-surfaces. The preferred method for imaging optics is always interferometry, which can perfectly limit the maximum slope that a surface may contain. The optical testing of MESSIER freeform primary mirror, proposed by the authors in Section-4.3, is an interesting setup and method for the freeform tests and verification after mirror’s fabrication.

6. On page-9, there are some typing errors in Table 2, and it seems to me that they should be like such as:

On-axis circular beam entrance diameter            356 mm

Focal length f                                                       890 mm

Focal-ratio                                                            f/2.5

Deviation angle 2i                                                22 degrees

In my opinion: This is a new idea of the freeform design and fabrication for space optics application. My recommendation is to accept this research paper as a novel optical design and fabrication for freeform optics in space application, after making a minor correction of typing errors in Table 2 (on page-9) as well as a bit of English error on page-4.

Submission Date

28 November 2018

Date of this review

11 Dec 2018 14:47:3

My answer to Referee-1

Your comments are convenient from the introduction up to point [5] included. For point [6] and the conclusion, my answer is as follow and for the editor :

In p.4, line 112 :  write « coefficients »  instead of  « coefficient s »

In p.9, Table 2, do not exactly follow the comment [6.] because some lines have been omitted. TO EDITOR : Redo Table 2 by including all the lines as they are in my original manuscript. There should be 11 lines in the table.

My own comment to the Editor

There is no Table 1. Please change the numbering; Table 2 shall become Table 1 and all other tables decreased by one unity. I am sorry about that.

In p.13, line 343, the sign before 1.224 is not adequate (in latex it is \simeq for similar-equal) (see our manuscript).

In line 344, the 2 location of signs ‘\in’ could be improved.

Line 376, write a convenient sign (as for line 344) before  –f  (the same as before 1.224 above). Same line, write 40 x 25

Reviewer 2 Report

Text:

This paper describes the optical design of a two mirror Schmidt telescope where the primary mirror is a freeform mirror.  A procedure for stressed optic polishing of an elliptical aperture freeform mirror is outlined, including modeling of the substrate to identify the deformation required to achieve the desired form. This methodology is then applied to the design of the proposed MESSIER telescope.

Several terms are used in non-standard ways:
Active optics: In astronomy both active and adaptive optics refer to optical adjustments made to the mirror figure during observations. Either slow changes to counteract the effects of a changing gravity vector (active optics), or, high cadence corrections to compensate for the blurring effects of the atmosphere (adaptive optics).  As this paper describes a method for polishing one of the telescope mirrors, which will then be used in a static configuration during observation, I do not feel that "active optics" is an appropriate way to refer to this particular optical design.  This is a freeform mirror produced by stressed optic polishing.

With this in mind, while the impact of active and adaptive optics to astronomy highlighed in lines 25-31 is true, I'm not sure it applies to this particular optical design.  I would then expect more references to other stressed optic polishing techniques.

The system is also described as a "three mirror anastigmat" or TMA.  However, only two of the mirrors are powered, M2 is included only to place the focal plane in a more convenient location.  Therefore I believe this system should be referred to as a "folded two mirror anastigmat".

It would be interesting to see a short discussion of the tolerances for this optical design. How well does the pressure need to be maintained during polishing?  How close to the prescription does the freeform optic need to be in order to maintain the required image quality?

Section 4.2 - did you investigate the impact of how the mirror is held during polishing? Does this modify the desired form of the substrate or internal pressure?

Lines 227-231 are a restatement of the introduction and could be removed.

line 233 It should be explicitly stated that the 32 mag/arcsec^2 and 37 mag/arcsec^2 are requirements flowed down from the MESSIER science requirements.  As it is written it seems like you are claming that just by being a "TMA" it will provide that sensitivity.  I did not understand that these were MESSIER science requirements until I read reference 5.  Adding some text like "MESSIER science goals require that any optical design deliver the following..." would help clarify.

line 255 "using a mirror as the first element" is redundant, I would rewrite as "In order to achive the best angular resolution over the FoV, the first element of a two mirror anastigmat should balance the meridian curvatures..."

Figures:
    1. This figure is not necessary, it does not add to the understanding of the paper.
    2. The 3rd sentence of caption repeats itself.
    3. This figure is confusing - would prefer y axis label equations to the left/right of numerical labels and all tic marks labeled (or at least min, midpoint, and max values).  References to 1st, 2nd, and bottom line are not immediately clear, are these the three lines in the upper plot, or the three lines of image spots?  Better to have all image spots together, above or below plot.  Perhaps better to make into Fig 3a and 3b. It looks like the 3rd row of spot images has a different x axis, this is misleading and should be called out in a more obvious way.  Are all spot sizes on the same scale?
    4. Text in image is hard to read, increase size or weight.
    5. Text in image is hard to read, increase size or weight.
    6. Consider including a second projection, the substrate form was not clear to me until seeing fig. 7.
    7. Uniform load label is misleading - I believe it is referencing the inlet port for the air pressure that causes the deformation?  In the figure it looks like the load is a point load applied off of the central axis. More clear in figure 13.
    10. Perhaps nice to add (y,z) axis label to figure to match with eq.14
    11. Perhaps nice to add (y,z) axis label to figure to match with eq.14
    12. Would be nice to have scale called out in arcsec (in caption is fine, i.e. box size is x.x arcsec)
    14. Either remove coordinate system in lower left corner, or make sure it is not cut off.  Would prefer color bar labels to be consistent in size, is there some significance to the change in font size?  Would be cleaner to redo the labels in microns, or only have significant digits with the label as [m x 10^-5] with space between the label and colorbar.
    15. Same general comments as Fig 14 regarding the color bar and labels.
    16. It looks like the "test mirror" got cut off of the edge of the figure (other mirrors in the figure are shown by a solid black line)

Tables:
    1. Referenced on line 375, but I do not see a table 1
    2. Contains missing line breaks, beam entrance and focal length labels are on the same line, f-ratio and deviation angle values are on the same line. Linear FoV should either be 20x40mm or 800mm^2.
    3. ok
    4. inner radii of elliptic cylinder: is this really 2ax x 2ay?  I read the figure as  2ax or 2ay is the diameter. I would also use a comma to separate similar measurements i.e. tx, ty = 18 mm, 18.33 mm.

Equations:
    1. I would prefer a horizontal line rather than a slash for the division sign.
    4. should this match the form of the y axis equation in figure 5 (rho^4 instead of rho^2 for the second term)?
    6. consider adding a reference for this equation
    7. consider adding a reference for this equation
    11. missing ) after v^2?

Author Response

ANSWER to Reviewer-2 (in the text in blue)

This paper describes the optical design of a two mirror Schmidt telescope where the primary mirror is a freeform mirror.  A procedure for stressed optic polishing of an elliptical aperture freeform mirror is outlined, including modeling of the substrate to identify the deformation required to achieve the desired form. This methodology is then applied to the design of the proposed MESSIER telescope.

Several terms are used in non-standard ways:
Active optics: In astronomy both active and adaptive optics refer to optical adjustments made to the mirror figure during observations. Either slow changes to counteract the effects of a changing gravity vector (active optics), or, high cadence corrections to compensate for the blurring effects of the atmosphere (adaptive optics).  As this paper describes a method for polishing one of the telescope mirrors, which will then be used in a static configuration during observation, I do not feel that "active optics" is an appropriate way to refer to this particular optical design.  This is a freeform mirror produced by stressed optic polishing.

With this in mind, while the impact of active and adaptive optics to astronomy highlighed in lines 25-31 is true, I'm not sure it applies to this particular optical design.  I would then expect more references to other stressed optic polishing techniques. For more references you can consult Astronomical Optics and Elasticity Theory – Actives Optics Methods, by Springer, 2009 (topic and some details on-line on Internet).

I use the terminology active optics for any deformable or deformed optics that requires elastic bending. As well in-situ stressing as stress polishing and elastic relaxation, and also for making mirrors or diffraction grating with replication technique on deformable matrices.

Three cases applies to active optics methods

1-Mirror polished in an unstressed state, then in situ bent by stressing (normal active optics in operation – small deformations – remote low time frequency).

2-Mirror or plate-lens polished spherical or flat under stress that becomes aspheric after stress release (normal stress polishing – large deformations).

3-Mirror submitted to a combination of both the 1) and 2) methods.

These methods present the fundamental advantage of generating high aspherics and freeform optics as smooth as those obtained by optical figuring of spheres and planes with full-size tools. Optical surfaces that are generated with full-size tools are free from high spatial frequency errors sometimes called ripple errors.

The system is also described as a "three mirror anastigmat" or TMA.  However, only two of the mirrors are powered, M2 is included only to place the focal plane in a more convenient location.  Therefore I believe this system should be referred to as a "folded two mirror anastigmat".

Line 14, replace ‘three-mirror-anastigmat (TMA)” by “folded-two-mirror-anastigmat or here called briefly three-mirror- anastigmat (TMA)”

Line 38, same modification as before.

It would be interesting to see a short discussion of the tolerances for this optical design. How well does the pressure need to be maintained during polishing? By a vacuum-pressure controler  How close to the prescription does the freeform optic need to be in order to maintain the required image quality? Provided by in-situ optical testing.

Section 4.2 - did you investigate the impact of how the mirror is held during polishing?  Foam elastic cushion. Does this modify the desired form of the substrate or internal pressure? Yes, but compensated by air pressure controller.

Lines 227-231 are a restatement of the introduction and could be removed. I prefer to keep it.

line 233 It should be explicitly stated that the 32 mag/arcsec^2 and 37 mag/arcsec^2 are requirements flowed down from the MESSIER science requirements.  As it is written it seems like you are claming that just by being a "TMA" it will provide that sensitivity.  I did not understand that these were MESSIER science requirements until I read reference 5.  Adding some text like "MESSIER science goals require that any optical design deliver the following..." would help clarify. All right :

Line 234, after “… brightness.”  add  "MESSIER science goals require that any optical design deliver a detection as..."

line 255 "using a mirror as the first element" is redundant, I would rewrite as "In order to achive the best angular resolution over the FoV, the first element of a two mirror anastigmat should balance the meridian curvatures..."

Line 255, replace "using a mirror as the first element…” by "using a mirror –instead of a refractive plate– as the first element…”

Figures:
    1. This figure is not necessary, it does not add to the understanding of the paper. Why not a bit of history for Messier’s portrait corresponding to project name ?
    2. The 3rd sentence of caption repeats itself.

Line 82, at end, suppress “circular”
    3. This figure is confusing - would prefer y axis label equations to the left/right of numerical labels and all tic marks labeled (or at least min, midpoint, and max values).  References to 1st, 2nd, and bottom line are not immediately clear, are these the three lines in the upper plot, or the three lines of image spots? x and y axes would not bring more understanding, The elements of the caption are well identified inside and both vertical scales.  Better to have all image spots together, above or below plot.  Perhaps better to make into Fig 3a and 3b. It looks like the 3rd row of spot images has a different x axis, this is misleading and should be called out in a more obvious way.  Are all spot sizes on the same scale? I agree for the scales spot images :

Line 106, add at the end “The three rows of spot images are at same scale”.
    4. Text in image is hard to read, increase size or weight. It is readable (editor).
    5. Text in image is hard to read, increase size or weight. It is readable (editor).
    6. Consider including a second projection, the substrate form was not clear to me until seeing fig. 7. The caption is conveniently  explained and Fig.7 adds to comprehension.
    7. Uniform load label is misleading [it is represented q] - I believe it is referencing the inlet port for the air pressure that causes the deformation?  In the figure it looks like the load is a point load applied off of the central axis. More clear in figure 13. Changing hereafter :

Line 193, change “…applied inside…” into “…applied all inside…”
    10. Perhaps nice to add (y,z) axis label to figure to match with eq.14. I mentioned in Section 4 that (y. z) is the telescope symmetry plane (whilst in section 2 and 3, the symmetry plane is (x, z). This is due to different optimization codes we used). However :

Line 271, replace “where now (y, x) is …” by “where now with Zemax code (y, x) is …”
    11. Perhaps nice to add (y,z) axis label to figure to match with eq.14. Same comment a above.
    12. Would be nice to have scale called out in arcsec (in caption is fine, i.e. box size is x.x arcsec). The correspondence is indicated just below the caption at line 281.
    14. Either remove coordinate system in lower left corner, or make sure it is not cut off.  Would prefer color bar labels to be consistent in size, is there some significance to the change in font size?  Would be cleaner to redo the labels in microns, or only have significant digits with the label as [m x 10^-5] with space between the label and colorbar. Adding digits will not provide more accuracy because of some uncertain accuracy in the elasticity modulus. And taking into account the small load pressure due to the polishing tool own weight and the small extra-load during polishing is necessary to obtain more accuracy. However this will be achieved in practice after 3 or 4 polishing iterations from the result of optical testing and the adjustment of the air pressure controller of resolution 5 x 10^-4.
    15. Same general comments as Fig 14 regarding the color bar and labels. This is just to make sure that we are far from the ultimate strength of the material (epoxy resin is more resistant to breakage than Zerodur materiel).
    16. It looks like the "test mirror" got cut off of the edge of the figure (other mirrors in the figure are shown by a solid black line). Of course the optical clear aperture of mirror M1 is smaller than that of the lens for testing; and this lens can also see part of the outer area of M1 which is polished but not used optically : it then prevent from the outside round corner ripple due to well know edge effect of polishing with the full-size tool. All this is well known.

Tables:
    1. Referenced on line 375, but I do not see a table 1. Yes, there is no Table 1. In fact, all numberings of the tables have to be updated :

Table 2 has to be renamed Table 1.  Table 3 becoming Table 2, etc.
    2. Contains missing line breaks, beam entrance and focal length labels are on the same line, f-ratio and deviation angle values are on the same line. Linear FoV should either be 20x40mm or 800mm^2.

Table 2 shall contain 11 lines. To Editor: Please recover my manuscript to redo this table.
    3. ok
    4. inner radii of elliptic cylinder: is this really 2ax x 2ay?  I read the figure as  2ax or 2ay is the diameter. Yes. I would also use a comma to separate similar measurements i.e. tx, ty = 18 mm, 18.33 mm. Yes

Table 4 after line 316, 5th line : change “radii” into “diameter”.

Table 4 after line 316, 6th line : change \times into commas as “tx, ty = 18 mm, 18.33 mm”

Equations:
    1. I would prefer a horizontal line rather than a slash for the division sign. Ok as it is.
    4. should this match the form of the y axis equation in figure 5 (rho^4 instead of rho^2 for the second term)? Yes.

Write rho^4 instead of rho^2 for the second term (as in my manuscript).
    6. consider adding a reference for this equation. Yes :

Line 155, add a the end [4][7]
    7. consider adding a reference for this equation. Yes :

Line 156, add a the end [4][7]
    11. missing ) after v^2?. Yes :

At numerator add a parenthesis after cos i as   2( … cos i). Also change reference (11) by (13).

Submission Date

28 November 2018

Date of this review

07 Dec 2018 21:05:02